# A Novel *Streptococcus thermophilus* FUA329 Isolated from Human Breast Milk Capable of Producing Urolithin A from Ellagic Acid

**DOI:** 10.3390/foods11203280

**Published:** 2022-10-20

**Authors:** Qitong Liu, Shu Liu, Qinwen Ye, Xiaoyue Hou, Guang Yang, Jing Lu, Yang Hai, Juan Shen, Yaowei Fang

**Affiliations:** 1Co-Innovation Center of Jiangsu Marine Bio-Industry Technology, Jiangsu Ocean University, Lianyungang 222005, China; 2Jiangsu Key Laboratory of Marine Bioresources and Environment, Jiangsu Ocean University, Lianyungang 222005, China; 3College of Food Science and Engineering, Jiangsu Ocean University, Lianyungang 222005, China; 4Key Laboratory of Marine Drugs, The Ministry of Education of China, School of Medicine and Pharmacy, Ocean University of China, Qingdao 266003, China; 5College of Food Science and Technology, Nanjing Agricultural University, Nanjing, 210095, China

**Keywords:** human breast milk, urolithin A, *Streptococcus thermophilus*, ellagic acid, biotransformation

## Abstract

Urolithin A, a metabolite of ellagic acid, has many beneficial biological activities for people. Strains capable of producing urolithin A from ellagic acid have the hope of becoming the next-generation probiotics. However, only a few species of these strains have been reported. In this study, FUA329, a strain capable of converting ellagic acid to urolithin A in vitro, was isolated from the breast milk of healthy Chinese women. The results of morphological observation, physiological and biochemical tests, and 16S rRNA gene sequence analysis confirmed that the strain FUA329 was *Streptococcus thermophilus*. In addition, the *S. thermophilus* FUA329 growth phase is consistent with the degradation of ellagic acid, and urolithin A was produced in the stationary phase, with a maximum concentration of 7.38 μM at 50 h. The corresponding conversion efficiency of urolithin A from ellagic acid was 82%. In summary, *S. thermophilus* FUA329, a novel urolithin A-producing bacterium, would be useful for the industrial production of urolithin A and may be developed as a next-generation probiotic.

## 1. Introduction

Ellagic acid, a polyphenol, occurs predominantly as ellagitannins in fruits, vegetables, and some plant foods [1,2]. Ellagic acid has antioxidant, antimutagenic, and anticancer properties; but the low level of ellagic acid in tissue and plasma limits its bioavailability [3,4,5,6]. Fortunately, some of the unabsorbed ellagic acid is metabolized by the intestinal microbiota of animals or humans to produce urolithins, which mainly include urolithin A, urolithin B, and isourolithin A [7,8]. Compared with ellagic acid, urolithins are more easily absorbed, and urolithins show a broad range of biological activities, especially anticancer, anti-inflammatory, antiproliferative, and anti-aging properties [9,10,11,12,13,14,15]. Individual differences as well as age-related differences are thought to influence the bioconversion capacity of ellagic acid to urolithins [16]. Based on the different intestinal metabolites produced from ellagitannins- or ellagic acid-rich foods, people are classified as three metabotypes: Urolithin metabotype A (UM-A), Urolithin metabotype B (UM-B), and Urolithin metabotype 0 (UM-0). The individuals of UM-A metabolize ellagic acid to only produce urolithin A, while the individuals of UM-B mainly produce isourolithin A and urolithin B, and urolithin A in low concentration. UM-0 is a group of people who cannot metabolize ellagic acid to generate urolithins [8].

Interestingly, urolithin A has more physiological functions than urolithin B and isourolithin A. For example, in adipocytes and hepatocytes, urolithin A significantly reduces triglyceride accumulation and increases energy expenditure [17,18]. In addition, urolithin A glucuronide exhibits a stronger anti-inflammatory activity than urolithin B glucuronide [19]. Moreover, urolithin A stimulates mitophagy, improves muscle health in both mice and humans, and extends the lifespan of the nematode *Caenorhabditis elegans* [20,21]. More importantly, urolithin A was assigned the status “generally recognized as safe (GRAS)” for use as an edible substance by the U.S. Food and Drug Administration in 2018. Consequently, urolithin A will be a future-oriented diet therapy and functional food. Amazentis launched high-purity urolithin A as a nutritional product that has anti-aging properties and promotes health.

Several studies have shown that gut microbiota plays a crucial role in the bioconversion of ellagic acid to urolithins, and consecutive intake of ellagic acid could enhance the capacity of certain species (especially among UM-0 and UM-B people) to increase urolithin A production, but the transformation mechanism and the gut microbiota involved in the transformation are unclear [22,23]. Only a limited number of studies have been conducted on the microbial species capable of transforming ellagic acid into urolithins. *Ellagibacter isourolithifaciens* DSM104140^T^ (*E. isourolithifaciens* DSM104140^T^) [24,25] and *Gordonibacter urolithinfaciens* DSM 27213^T^ (*G. urolithinfaciens* DSM 27213^T^) [26,27,28] were confirmed to convert ellagic acid to isourolithin A and urolithin C. Only a single strain of *Bifidobacterium pseudocatenulatum* INIA P815 (*B. pseudocatenulatum* INIA P815) could produce urolithin A [29]. Meanwhile, until now, urolithin A could only be synthesized using the chemical method [30]. Therefore, the isolation and identification of microbial species capable of transforming ellagic acid into urolithins, especially GRAS strains, have gained increasing attention from researchers because microbial species could be used not only to study the biotransformation mechanism of urolithin A, but also to develop probiotics and functional foods.

The biosynthesis of urolithin A has significant economic benefits; therefore, it is necessary to screen strains with the ability to biosynthesize urolithin A. Aging plays a crucial role in ellagic acid metabolism in humans, and the population of UM-A in infants accounts for the highest proportion (>80%). Several urolithins can be transported to infants through breast milk, and lactic acid bacteria in breast milk also allow for the establishment of gut microbiota in infants [16,31,32]. *Bifidobacterium* species are an advantageous flora in infants’ guts [33], and the only urolithin A- producing strain of *Bifidobacterium pseudocatenulatum* INIA P815 belongs to the *Bifidobacteriaceae* family. Thus, we speculated that the GRAS strains capable of transforming ellagic acid into urolithins could be isolated from human breast milk.

In this study, a urolithin A-producing *S. thermophilus* FUA329 strain was successfully screened from the breast milk of healthy Chinese women. Due to *S. thermophilus* having been conferred with the GRAS status in the United States and the qualified presumption of safety (QPS) status in the European Union [34], and the breast milk sources used to screen strains are safer than feces, FUA329 is more likely to have high safety. Then, the time course of transformation of ellagic acid to urolithin A by the FUA329 strain was also determined through high-performance liquid chromatography (HPLC) and high-performance liquid chromatography-tandem mass spectrometry (HPLC-MS/MS). The results showed that the FUA329 corresponding conversion efficiency of urolithin A from ellagic acid was 82%. Therefore, human breast milk can be used to screen ellagic acid metabolizing strains. Furthermore, *S. thermophilus* FUA329 could be also applied to investigate the molecular mechanism of transformation of ellagic acid to urolithin A.

## 2. Materials and Methods

### 2.1. Chemicals

*L*-cysteine hydrochloride, urolithin A, isourolithin A, urolithin D, urolithin E, urolithin M5, and urolithin M6 were purchased from Dalton Pharma Services (Toronto, Canada). Urolithin B and urolithin C were purchased from Shanghai ZZBIO Co., Ltd. (Shanghai, China). Ellagic acid was purchased from Sigma-Aldrich (St. Louis, MO, USA). Mass spectrometry (MS) or liquid chromatography (LC)-grade solvents methanol, acetonitrile, formic acid, and ethanol were purchased from Merck (Darmstadt, Germany).

### 2.2. Collection of Breast Milk from the Healthy Women

Before the experiment started, all volunteers signed an experimental informed consent form at the Jiangsu Ocean University (Lianyungang, China) and provided breast milk samples. The whole process of the experiment did not involve human experiments and had no harm to the human body, not involving ethics.

All volunteers signed an experimental informed consent form at the Jiangsu Ocean University (Lianyungang, China) and provided breast milk samples. In total, 7 samples were collected from 7 healthy lactating women (age: 25–37 years old) by using a sterile breast pump after thorough cleaning of the whole breast with sterile water three times. The samples were kept in an ice box and sent within half an hour to the lab, and then used for the fermentation experiments.

### 2.3. Fermentation Medium

Anaerobe Basal Broth (ABB) was purchased from Oxoid (Basingstoke, Hampshire, England). An amount of 80 mL medium aliquots was dispensed in the 100 mL anaerobic bottles. Then, the bottles were autoclaved at 121 °C for 15 min and allowed to cool to room temperature in an anaerobic chamber (Don Whitley Scientific, West Yorkshire, UK) under an atmosphere of N_2_/H_2_/CO_2_ (80:10:10).

### 2.4. Screening of Human Milk Samples for Conversion of Ellagic Acid to Urolithin A In Vitro

Under sterile conditions, 1 mL of the human breast milk sample was inoculated into the aforementioned ABB medium supplemented with 20 μM ellagic acid and 0.05% *L*-cysteine hydrochloride. Then, the mixture was cultured for 4 days in the anaerobic chamber.

In addition, the human breast milk-free medium containing ellagic acid and the ellagic acid-free medium containing human breast milk were used as the control groups. The samples (10 mL) were collected once per day from each culture and extracted with a 10 mL organic solvent (C_2_H_3_N:H_2_O:CH_2_O_2_ 80:19.9:0.1). Then, the extracts were analyzed by HPLC (1260 Series, Agilent Technologies, Germany) and HPLC-MS/MS (Thermo Fisher, Waltham, MA, USA) analysis. Six urolithin standards (20 μM) were analyzed by HPLC at 305 nm. If urolithin A was detected in the fermentation broth, the corresponding breast milk sample was used to screen the urolithin A-producing bacteria.

### 2.5. Screening Urolithin A-Producing Bacteria

A sterile distilled ABB medium was used to dilute the fermentation broth containing urolithin A to 10^−3^–10^−7^. Then, 0.1 mL of the diluted fermentation broth was spread on the ABB agar plate, and 12 plates were incubated at 37 °C in the anaerobic chamber under the atmosphere of N_2_/H_2_/CO_2_ (80:10:10). All colonies were separately inoculated in the 10 mL ABB containing 20 μM ellagic acid. After anaerobic incubation at 37 °C for 5 days, the broth (9 mL) was separately extracted with 10 mL C_2_H_3_N:H_2_O:CH_2_O_2_ (80:19.9:0.1, *v*:*v*:*v*) and further analyzed through HPLC and HPLC-MS/MS analysis. Urolithin A-producing colonies were stored in an ABB medium with 20% (*v*/*v*) glycerol at −80 °C for further study.

### 2.6. Identification of the Strain FUA329

Phenotype identification of the strain FUA329 was made based on colony morphology and gram staining, and then its biochemical and physiological characteristics were measured based on Bergey’s Manual of Systematic Bacteriology [35].

Total genomic DNA was extracted using a commercial DNA extraction kit (Takara, Dalian, China). Using the total genomic DNA as a template and the primers 27f and 1492r, the 16S rRNA was amplified according to the procedures described by Flavio Tidona [36]. The PCR products were purified using an OMEGA Gel Extraction Kit, and then sent to Shanghai Sangon Biological Engineering Co., Ltd. (Shanghai, China) for sequencing. The similarity of 16S rRNA sequences was compared using the BLAST search in NCBI. Phylogenetic analysis of the strain FUA329 based on the 16S rRNA sequence was performed using MEGA 11.0.

### 2.7. Time Course of the Transformation of Ellagic Acid to Urolithin A

*S. thermophilus* FUA329 in the frozen glycerol stock was streaked on the ABB agar plate and anaerobically incubated at 37 °C for 2 days. A single colony was picked from the plate and inoculated in 5 mL ABB at 37 °C for 5 days. An amount of 2 mL of the diluted inoculum was transferred to 150 mL ABB to obtain a final concentration of 1.5 × 10^4^ CFU·mL^−^^1^ and incubated under anaerobic conditions consisting of N_2_/H_2_/CO_2_ (80:10:10, *v*:*v*:*v*) at 37 °C for 7 days. Ellagic acid was added to ABB with a final concentration of 20 μM. Then, 5 mL of the sample was collected at appropriate time intervals during the 7-day incubation and extracted with the same volume of C_2_H_3_N:H_2_O:CH_2_O_2_ (80:19.9:0.1, *v*:*v*:*v*). The extracts were then detected using HPLC analysis and HPLC-MS/MS analysis.

### 2.8. HPLC Analysis

Chromatographic separation was performed on a ZORBAX SB-C18 column (250 × 4.6 mm, 5.0 μm) (Agilent Technologies, Santa Clara, CA, USA). One percent methanol (A) and acetonitrile (B) were used as the mobile phases. The flow rate was 1.0 mL/min, and the injection volume was 5 µL. UV chromatograms were taken at 305 nm. The gradient profile was 0~15 min, 0~20% solvent B; 15~20 min, 20~70% solvent B; 20~21 min, 70~95% solvent B; 21~24 min, 95~100% solvent B; 24~25 min, 100~20% solvent B.

### 2.9. HPLC-MS/MS Analyses

HPLC-MS/MS (Thermo Fisher) was used to analyze urolithin A at 305 nm by using a ZORBAX SB-C18 column (250 × 4.6 mm, 5.0 μm) (Agilent Technologies, Santa Clara, CA, USA) and HR-ESI-MS (high-resolution electrospray ionization mass spectroscopy) can range from 150.0 to 2000.0 Da. Sheath gas-flow rate, 45 arb; aux gas-flow rate, 15 arb; capillary temp: 320 °C; and S-Lens RF Level 50 T. The column temperature was 30 °C. The mobile phase consisted of solvent A (0.1% formic acid water) and solvent B (acetonitrile). The gradient elution was as follows: 0~10.5 min, 10~100% solvent B; 10.5~12.5 min, 100% solvent B; 12.5~12.6 min, 100%~10% solvent B; 12.6~16.5 min, 10% solvent B for re-equilibration. Urolithin A in the fermentation broth was first identified through HPLC and further identified by comparing the molecular mass of the compound obtained with that of a pure standard by using HPLC-MS/MS.

## 3. Results

### 3.1. Screening of Human Breast Milk Samples Converting Ellagic Acid to Urolithin A In Vitro

In three parallel experiments, the HPLC analysis showed that urolithin A was produced by one human breast milk sample (Figure 1b). To accurately evaluate the type of urolithin, we designed a standard diagram including six major urolithin standards and ellagic acid standards with a concentration of 20 μM (Figure 1a). As shown in Figure 1b, urolithin A could be detected in the breast milk of volunteer seven by comparing it with the standard diagram (Figure 1a). Thus, the urolithin A-producing bacterium was isolated in human breast milk, which revealed that human breast milk can be used to screen ellagic acid metabolizing strains.

### 3.2. Screening Urolithin A-Producing Bacteria

In this study, 93 strains were obtained from the breast milk of volunteer 7. To further identify whether strain FUA329 could transform ellagic acid to urolithin A in vitro, urolithin A in the fermentation broth was first identified through HPLC and further identified by comparing the molecular mass of the compound obtained with that of a pure standard by using HPLC-MS/MS. The HPLC and HPLC-MS/MS results revealed that FUA329 was the only strain that could convert ellagic acid to urolithin A (Figure 2 and Appendix A).

### 3.3. Identification of the Urolithin A-Producing Bacteria

The strain FUA329 on ABB agar plates was oyster white, smooth, and eminent. Under the microscope, strain FUA329 was coccus-shaped and gram-positive, with most cells arranged as chains (Appendix A). Based on the results of the physiological and biochemical characteristics of strain FUA329 (Table 1), strain FUA329 was preliminarily classified as *S. thermophilus*.

The 16S rRNA sequence of strain FUA329 was submitted to GenBank with accession number OM892001. Up to March 17, 2022, the genomes of 184 *S. thermophilus* strains have been submitted to the NCBI database. This strain is also available at the China General Microbiological Culture Collection Center (CGMCC no. 24963). The phylogenetic tree suggested that strain FUA329 belongs to the genus *Streptococcus* (Figure 3). The closest relative of this strain was strain DSM 20617^T^. The 16S rRNA gene sequence of FUA329 showed 99.57% identity to the sequence of the *S. thermophilus*. Thus, strain FUA329 was identified as *S. thermophilus*. *S. thermophilus*, which is listed in the edible strain catalog, is one of the most widely used lactic acid bacteria in dairy applications and has the GRAS and QPS status. Therefore, *S. thermophilus* FUA329 could be a promising probiotic candidate.

### 3.4. Time Course of Metabolizing Ellagic Acid to Urolithin A by Strain FUA329

To further understand the relationship between the growth characteristics of FUA329 and ellagic acid metabolism, we determined the growth curve of FUA329 and the time course of metabolizing ellagic acid to urolithin A. As shown in Figure 4, the *S. thermophilus* FUA329 growth phase was less consistent with the degradation of ellagic acid. Urolithin A formation occurred during the stationary phase of *S. thermophilus* FUA329 growth. The metabolite urolithin A (dihydroxy-urolithin) was observed at 44 h after incubating ellagic acid with *S. thermophilus* FUA329. The concentration of FUA329 peaked at 50 h (maximum concentration: 7.38 μM) and then decreased (Figure 4). A large proportion of the ellagic acid supplied (12 μM) was finally converted into urolithin A. Although some ellagic acid remained unmetabolized in the medium (1.3 μM), the corresponding conversion efficiency of urolithin A was 82% at 50 h (Table 2).

## 4. Discussion

Ellagic acid is a natural polyphenol that mainly exists in some natural dietary sources, such as berries (pomegranate, strawberry, and raspberry), teas, and nuts (walnut and chestnut) [23,37]. There is a correlation between ellagic acid and health effects, but it is difficult for humans to absorb because of the low bioavailability of ellagic acid [6]. However, recent studies have found that ellagic acid can be metabolized into urolithins through intestinal microbiota in the colon [38,39]. Unlike ellagic acid, urolithins are more easily absorbed in colonic mucosa and undergo phase II metabolism (methylation, glucuronic acid, and sulfonation) by human cellular enzymes to produce gluconic anhydride and sulfate conjugates [40]. A large amount of experimental data shows that urolithins are the active component of ellagic acid in vivo. Urolithins play an important role in regulating processes, such as anti-oxidation, anti-inflammation, bacteriostasis, anti-cancer, anti-obesity, anti-aging, and estrogen receptors [13,41,42,43]. Thus, the pressing research topics in the field of urolithins have focused on metabolism, microbiota, cell, mechanism, expressions, and pathways in recent years. Recent studies of the biological activity of urolithins were based on in vitro studies, and their bioavailability and tissue distribution were rarely considered. However, to obtain more urolithin-producing strains, the distribution of urolithins in different biological compartments should be studied. Most of the previous studies about the distribution of urolithins focus on plasma, urine, and feces in humans or other human tissues in vivo, and only several aglycone and conjugate urolithins were detected in animals, such as ruminal fluids of bulls (Table 3). To date, urolithin A could only be detected in tissues or fluids related to the intestinal tract, such as urine and feces, however, these tissues or fluids contain more pathogenic bacteria. In addition, only scarce amounts of urolithin A could be detected in the rest samples of tissues or body fluid, such as plasma, brain, and bile.

According to the type of urolithin metabolism, people are divided into three types: metabolotype A, metabolotype B, and metabolotype 0. The individuals in metabotype A could metabolize ellagic acid to produce urolithin A, while the individuals in metabotype B produce urolithin A, isourolithin A and/or urolithin B. Metabotype 0 represents the population that could not produce urolithins [8]. However, urolithin metabolic type can be affected by region and age. Thus, the probability of screening urolithin A- producing strains from human milk is also affected by region and age. There are higher percentages of UM-A in younger people [16]. Thus, it is more probable that urolithin A- producing strains are from adolescents, and even infants. Maternal gut microbiota have a great influence on the infant’s gut during pregnancy and lactation. Thus, in this study, three replication experiments were performed, and the results showed that human breast milk can be used to screen ellagic acid metabolizing strains. Because of intimate contact during childbirth, nursing, and breastfeeding, the greatest contribution to an infant’s gut microbiome development may be the mother as one of the external factors [51]. The delivery mode has a great impact on the early intestinal microbe colonization in infants [52], breastfeeding also has a significant effect on the development of an infant’s gut microbes. Several studies have found that breastfed babies have higher levels of *Lactobacillus* [52,53]. This finding indirectly showed that breast milk is rich in *Lactobacillales* or lactic acid bacteria (LAB). Several pilot studies demonstrated that ellagic acid and its metabolites like urolithins are absorbed by the nursing infant from breast milk, excreted in urine, and impact the infant gut microbiome. The concentration of urolithins in breast milk increased over time. Phenolic compounds in breast milk could be a way to promote neuroprotective, antioxidant, and anti-inflammatory health benefits in infants [50]. In this research, we found that human breast milk could be a novel source for screening urolithin A-producing strains, and it is safer than using feces or urine as a source. Human breast milk establishes the foundation of infants’ gut microbiota and is reported to be a key source of probiotic lactic acid bacteria [37]. Thus, the gut microbiota of the vast majority of infants is capable of transforming ellagic acid to urolithins [54,55]. Human breast milk could be a potential resource for isolating urolithin-producing strains, as confirmed by the present study results [55].

Some intestinal microbiota have shown the ability to transform ellagic acid to urolithin A in vivo [42]. However, these strains or flora are not identified, including the strains or flora from human breast milk. Moreover, although urolithin A was discovered in humans nearly 40 years ago, enzymes responsible for metabolizing ellagic acid to produce urolithins have remained unknown [56,57]. Therefore, it is necessary to isolate strains capable of producing urolithin A from ellagic acid for investigating the molecular mechanism of the conversion of ellagic acid to urolithins. To date, four urolithins-producing strains have been reported, and the ellagic acid biotransformation processes of these strains differed from each other. *Gordonibacter* was the first reported genus capable of converting ellagic acid to urolithin M5, urolithin M6, and urolithin C [28,58]. Ellagic acid catabolism and urolithin formation occurs during the stationary growth phase of *G. urolithinfaciens* and *G. pamelaeae* species, and urolithin C was the end product of ellagic acid catabolism by the two strains [27,28]. For the strain *E. isourolithinifaciens,* ellagic acid catabolism and urolithin C accumulating also occurred during the stationary growth phase, but isourolithin A was the end product of ellagic acid catabolism [24,30]. *B. pseudocatenulatum* INIA P815 could rapidly metabolize ellagic acid to urolithin A (dihydroxy-urolithin) and urolithin B (monohydroxy-urolithin) [29]. Similar results were also reported by García-Villalba et al. [8]. Among them, only *B. pseudocatenulatum* INIA P815 exhibited the ability to convert ellagic acid to urolithin A (Table 4). Moreover, all four strains are not listed in the edible strain catalog.

In this study, we screened the high-safety strains capable of biotransforming ellagic acid to urolithin A and found that urolithin A formation occurred during the stationary phase of *S. thermophilus* FUA329 growth. There are some strengths of the strain FUA329. The source of FUA329 was human breast milk, which was safer. Moreover, the results of the 16S rRNA gene sequences and genome sequences indicated that this bacterium belonged to *S. thermophilus*, which has been recognized as the safe strain in the edible strain catalog, granted the GRAS and QPS status [40]. Another urolithin A- producing strain INIA P815 screened from fecal samples belongs to *B. pseudocatenulatum*, but its safety is unknown [29]. In addition, the strain FUA329 of the corresponding conversion efficiency of urolithin A from ellagic acid was 82%, but the conversion rate of urolithin A regarding INIA P815 hasn’t been reported. In the study of Rocío et al. [59], ellagic acid was 62% converted to urolithins at 36 h in the human fecal culture of urolithin A production. In contrast, the lack of growth of *Bifidobacterium* in isourolithin A production could indicate that this microbial group is not involved in urolithin production. Thus, the strain FUA329 is a safe strain and has the highest conversion rate of urolithin A thus far.

However, the safety and probiotic characteristics of *S. thermophilus* FUA329 haven’t been supported in this research. Thus, we will assess the safety and probiotic characteristics of FUA329 in a future study, with plans to develop it as a next-generation probiotic and ferment yogurt containing ellagic acid. Presently, we know the two urolithin A- producing strains from different bacterial genera, the *Bifidobacteriaceae* family and *Streptococcaceae* family, and we can further explore the connection and mechanisms related to producing urolithin between them in the future.

## 5. Conclusions

In this study, urolithin A was produced from ellagic acid by a single bacterial strain from human breast milk. Furthermore, this is the first urolithin A-producing bacterial strain present in the edible strain catalog. Therefore, the strain *S. thermophilus* FUA329 could potentially be developed as a next-generation probiotic. *S. thermophilus* FUA329 could also be used to explore the pathway of metabolizing ellagic acid to urolithin A and even used for the production of urolithin A at an industrial scale to save costs because of its high conversion efficiency. However, the metabolic pathways of FUA329 using ellagic acid as the substrate to produce urolithin A require further study.

## Figures and Tables

**Figure 1 foods-11-03280-f001:**
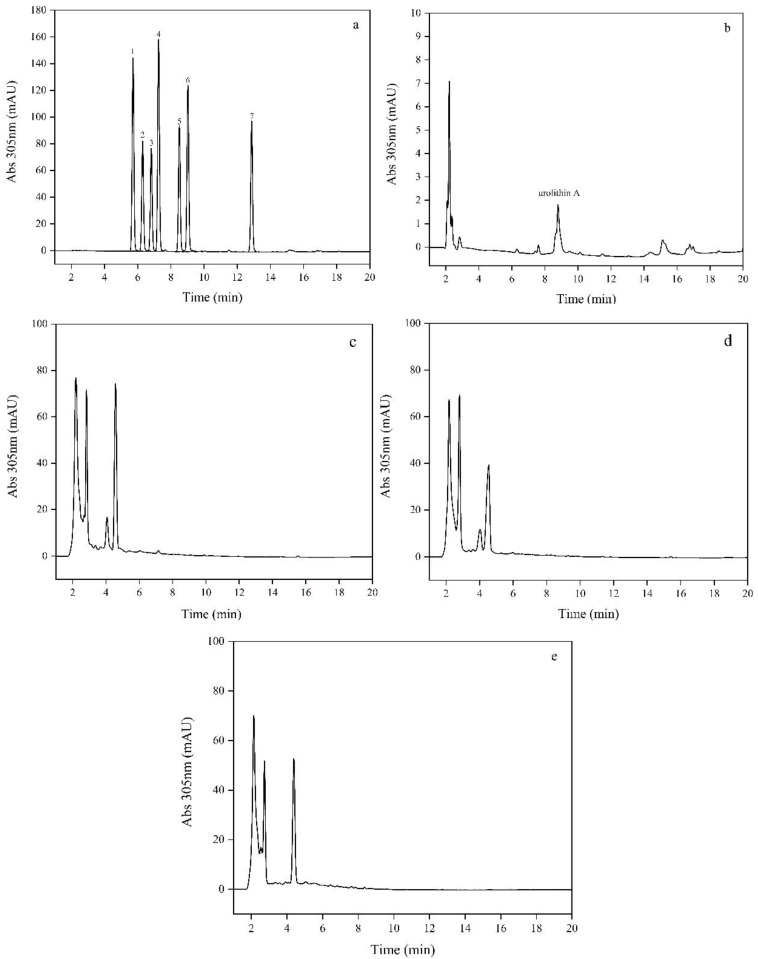
HPLC analysis at 305 nm of samples. (**a**) Standard of urolithins (20 μM): urolithin D (1), ellagic acid (2), urolithin M6 (3), urolithin C (4), isourolithin A (5), urolithin A (6), and urolithin B (7); (**b**) breast milk sample of volunteer 7; (**c**) medium sample without breast milk; (**d**) medium sample without ellagic acid; and (**e**) one of the breast milk samples of non-urolithin A.

**Figure 2 foods-11-03280-f002:**
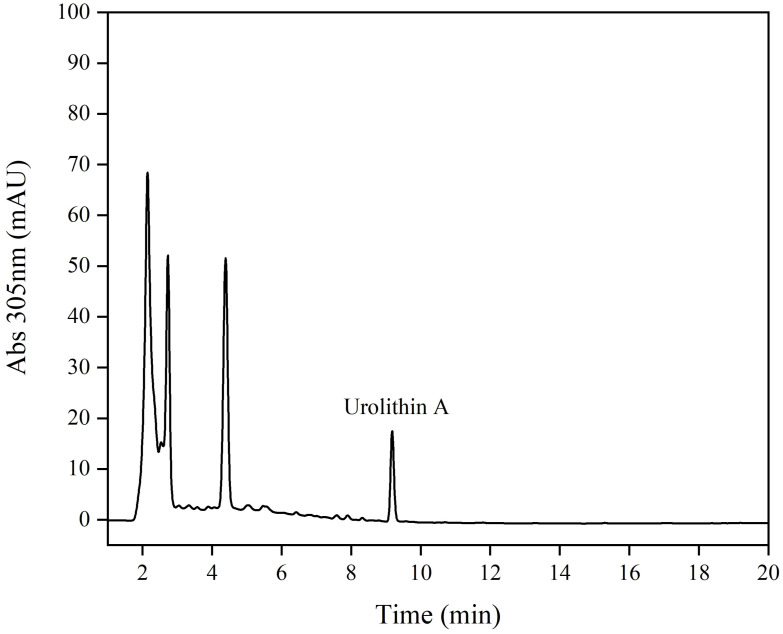
HPLC analysis of the fermentation broth of the strain FUA329.

**Figure 3 foods-11-03280-f003:**
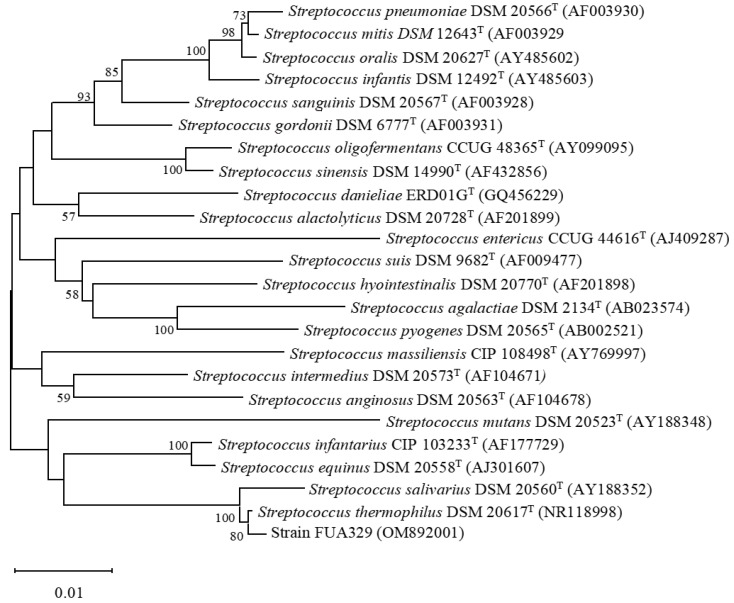
Phylogenetic tree of the strain FUA329 within the genus Streptococcus based on 16S rRNA sequences. The branching pattern was generated by using the maximum-likelihood method. Bootstrap values over 50% from 1000 replications are shown on the branches. The bar below the tree represents 1 nucleotide change per 1000 nucleotides.

**Figure 4 foods-11-03280-f004:**
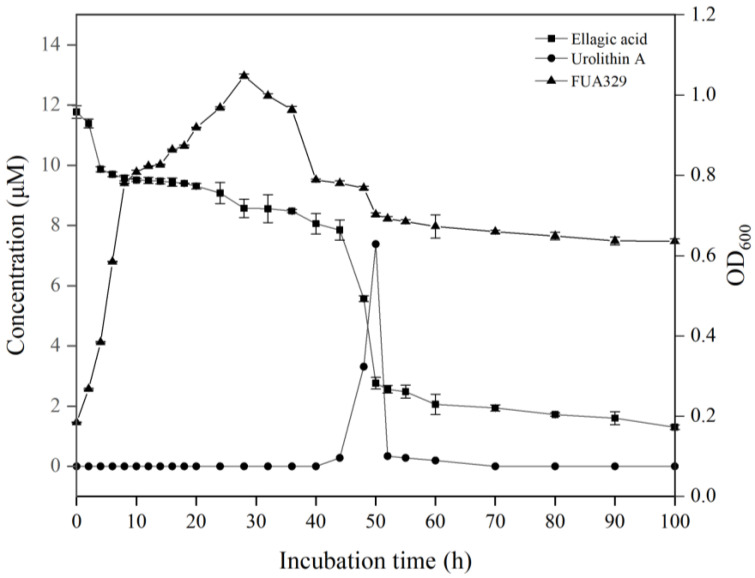
Time course of ellagic acid catabolism and urolithin A accumulating in the fermentation broth of the strain FUA329. Ellagic acid (solid squares), urolithin A (solid circles), and the strain FUA329 growth curve (solid triangles).

**Table 1 foods-11-03280-t001:** Morphological, physiological, and biochemical characteristics of the strain FUA329.

Properties	Result
Shape	Coccus
Gram stain	+
Motility	−
10 °C	−
45 °C	−
3% NaCl	−
pH 9.0	−
60 °C for 30 min	−
Catalase	−
H_2_S production	−
Indole	−
Nitrate reduction +	−
Methyl red test	−
Vopes–Proskauer test	+
Gelatin liquefaction	−
Esculin	−
Starch	+
Arabinose	−
Cellobiose	−
Galactose	−
Glucose	+
Lactose	+
Sucrose	+
Fructose	+
Maltose	−
Mannitol	−
Melicitose	−
Raffinose	−
Mannose	−
Melibiose	−
Rhamnose	−
Ribose	−
Sorbitol	−
Fucose	−
Xylose	−
Synanthrin	−

+, positive results; −, negative results.

**Table 2 foods-11-03280-t002:** The key time point of the strain FUA329 growth and urolithin A production.

Time/h	Concentration of Ellagic Acid (μM)	Concentration of Urolithin A (μM)	OD600 Value of Strain FUA329
0	11.78 ± 0.21	0	0.185 ± 0.001
12	9.51 ± 0.1	0	0.823 ± 0.001
28	8.6 ± 0.31	0	1.047 ± 0.005
48	5.56 ± 0.1	3.31 ± 0.1	0.768 ± 0.004
50	2.77 ± 0.19	7.38 ± 0.07	0.702 ± 0.004
52	2.57 ± 0.13	0.34 ± 0.02	0.692 ± 0.005
80	1.72 ± 0.06	0	0.649 ± 0.009
100	1.33 ± 0.08	0	0.636 ± 0.006

**Table 3 foods-11-03280-t003:** Distribution of urolithin A and its conjugate metabolites in biological fluids and tissues.

	Administration	Tissue/Biological Fluid	Identified Metabolites	Refs.
Animals				
Rats	pomegranate extract	kidney, liver	urolithin A glucuronide	[44]
	oak-flavored milk powder	urine (U), plasma (P), feces (F)	urolithin A (F), urolithin A glucuronide (U)	[45]
Pigs	fresh acorns	P, U, F, bile (Bi), intestinal lumen (LUM), gastrointestinal tissue (GIT)	urolithin A (U, F, Bi, LUM, GIT), urolithin A glucuronide (Bi, P, GIT)	[46]
Bulls	hay and oak leaves	ruminal fluid (RF), F, U, P	urolithin A (RF, F), urolithin A glucuronide (U), urolithin A sulfate (P)	[47]
Humans				
Healthy adult volunteers	walnuts	F	urolithin A	[7]
	black tea	U	urolithin A 3-*O*-glucuronide, urolithin A 8-*O*-glucuronide	[48]
	pomegranate extract	P	urolithin A, urolithin A glucuronide	[49]
Infants	Pomegranate juice	F, U	urolithin A glucuronide (U)	[50]
Healthy lactating women	walnuts	Breast milk	urolithin A	

**Table 4 foods-11-03280-t004:** Urolithins and urolithins-producing strains [24,27,28,30,58].

Characteristic	Types	Structure	Metabolic Strains	Source of Strains
Tetrahydroxy	Urolithin M6	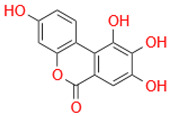	*G. urolithinfaciens* DSM 27213^T^ *G. pamelaeae* DSM 19378^T^	Fecal samples
			*E. isourolithinifaciens* DSM 104140^T^	
			*B. pseudocatenulatum* INIA P815	
	Urolithin D	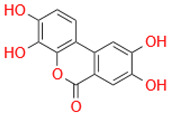	*B. pseudocatenulatum* INIA P815	
Trihydroxy	Urolithin C	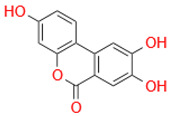	*G. urolithinfaciens* DSM 27213^T^ *G. pamelaeae* DSM 19378^T^	
			*E. isourolithinifaciens* DSM 104140^T^	
			*B. pseudocatenulatum* INIA P815	
Dihydroxy	Urolithin A	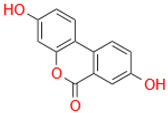	*B. pseudocatenulatum* INIA P815	
			*S. thermophilus* FUA329	Breast milk
	IsoUrolithin A	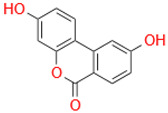	*E. isourolithinifaciens* DSM 104140^T^	Fecal samples
			*B. pseudocatenulatum INIA* P815	
Monohydroxy	Urolithin B	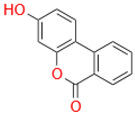	*B. pseudocatenulatum* INIA P815	

## Data Availability

The data presented in this study are available on request from the corresponding author.

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
