# Peer review of "A Novel Streptococcus thermophilus FUA329 Isolated from Human Breast Milk Capable of Producing Urolithin A from Ellagic Acid"

_foods, 2022, doi:10.3390/foods11203280_

Round 1
Reviewer 1 Report
The manuscript entitled "A novel Streptococcus thermophilus FUA329 isolated from human breast milk capable of producing urolithin A from ellagic acid" aims to isolate the strain FUA329 which converts ellagic acid into urolithin A in vitro from the breast milk of Chinese healthy women. Some major and minor questions/suggestions are stated below and should be addressed:
1. In the methods' section, the authors should include the tested concentrations of standards (particularly urolithin A and ellagic acid) regarding the calibration curves for HPLC analysis in the subsections 2.8 and 2.9, as well as a sentence about the results presentation.
2. Why were the target compounds and bacterium strain only detected in the breast milk from volunteer 7? This could lead to low reproducibility of the data obtained since the findings were just found in the breast milk of 1 volunteer from the 7 volunteers. In addition, how can the authors affirm with certainty that the strain FUA329 will be found in the breast milk of more women? Please explain it in detail and discuss it in the manuscript.
3. The intra- and inter-variability of the data should be also discuss in studies using samples from animals and humans. How can the results be affected by the intra- and inter-variability? Please explain it.
4. The discussion should be improved considering the main strengths and weaknesses of the study, as well as potential applications in food/nutraceutical industry. Also, the manuscript may benefit from a more detailed explanation and interpretation of the main findings of the study, as well as a comparison with the available literature about the subject on study.
5. What was the % of similarity of FUA329 strain with S. thermophilus bacterium? Please add this result in the manuscript (subsection 3.3).
6. The authors should present the concentrations of ellagic acid, FUA329 strain and urolithin A determined (subsection 3.4) in a table together with the standard deviation and provide the statistical analysis to understand if there are significant differences.
7. In Figure 6, please include the standard deviation. Also, correct the word "Concentration" in the axis y of the graph.
8. References need careful revision. Please format the references according to the journal guidelines.
Minor comments:
- Lines are missing on the right side margin of the manuscript. Please use the template of Foods journal to prepare the manuscript, following the journal guidelines and instructions. Additionally, format the manuscript according to the journal guidelines, with particular attention to tables and figures.
- The references are not cited properly throughout the manuscript. The authors should follow the numbered references' style. Please revise them in all manuscript according to the journal guidelines.
- The manuscript may benefit from a brief list of the most used abbreviations.
- The novelty of the study in comparison with the already available literature could be highlighted in the introduction, for example, to clarify the gap that the present study fulfills in food/nutraceutical research field.
- Figures and tables should be placed immediately after where they are first mentioned in the manuscript.
- Some minor comments are also stated in the pdf file uploaded with the revisions. Please see the file attached.

Author Response
Dear Reviewer,
We sincerely thank you for your valuable review that we have used to improve the quality of our manuscript. According to your comments, we have made extensive modifications to our manuscript. In this revised version, changes to our manuscript were all highlighted within the document by using red-colored text. And the detailed point-by-point responses are attached below. Please see the attachment.
Best regards,
Ms. Liu

Reviewer 2 Report
Liu et al addressed very important question in the polyphenolic research field, i.e., identifying bacteria responsible for urolithin A production. The MS is well-written and easy to follow.
Few changes and additional data will significantly improve quality of this MS.
1. Authors should show the negative results (i.e., milk data of non-UroA producers).
2. Similarly, exemplary of other bacteria that are not capable of producing UroA.
3. Again, authors state that S. thermophilus DSM 20617T (NR118998) has 99.5% 16S gene comparable with UroA producing strain FUA329. It will strengthen data, if authors show DSM 20617T (NR118998) failed (?) to produce UroA.
4. Since, pure strain FUA329 is available in hand, I failed to understand, what prevented the authors to sequence the full bacterial genome.
Author Response

(The authors gave the same response as above.)

Reviewer 3 Report
The manuscript reports a novel Streptococcus thermophilus FUA329 isolated from human breast milk capable of producing urolithin A from ellagic acid. The topic is interesting since it aims to explore novel bacterial strains (so far only a few strains have been investigated) capable of producing urolithin A from ellagic acid. The outcome is of interest since urolithin A, which is a metabolite of ellagic acid, does present many beneficial biological activities and might represent a next-generation probiotic. However, there are many aspects that need to be clarified. The main weak point if the lack of quantitative data which is crucial for this type of study.
English is acceptable although it can be further improved.
2.8. HPLC analysis and Section 2.9. HPLC-MS/MS analyses. It is not clear if the same instrument was employed. If not, what is the reason? The gradient profiles are not clear and need to be better described.
Section 2.9. HPLC-MS/MS analyses, the following lines should be transferred to the Results and Discussion section “To further identify whether strain FUA329 could transform ellagic acid to urolithin A in vitro, urolithin A in the fermentation broth was first identified through HPLC and further identified by comparing the molecular mass of the compound obtained with that of a pure standard by using LC-MS/MS”.
The Thermo Fisher UPLC system should be described in all detail.
Chromatograms in Figure 1B-D and 2 do not report any peak numbering.
Figure 3 can be moved to supplementary materials.
Author Response

(The authors gave the same response as above.)

Round 2
Reviewer 1 Report
The authors addressed the questions and made all the modifications required by the reviewer. Introduction and Methods sections were improved and more details about the methodologies were provided. The references were reformatted according to journal guidelines. Two tables were added to better explain the findings and rationale of the study. In my perspective, the paper is now in conditions to be considered for publication in Foods journal.
Author Response
Thank you for your genuine review comments again.
Reviewer 2 Report
Authors addressed my concerns.
Author Response

(The authors gave the same response as above.)

Reviewer 3 Report
The authors have adequately addressed all remarks and the improved paper can be now accepted in the present form
Author Response

(The authors gave the same response as above.)
